# Unchanged Fatality Rate on Austrian Ski Slopes during the COVID-19 Lockdown

**DOI:** 10.3390/ijerph19137771

**Published:** 2022-06-24

**Authors:** Markus Posch, Johannes Burtscher, Gerhard Ruedl, Elena Pocecco, Martin Burtscher

**Affiliations:** 1Department of Sport Science, University of Innsbruck, 6020 Innsbruck, Austria; markus.posch@uibk.ac.at (M.P.); gerhard.ruedl@uibk.ac.at (G.R.); elenapocecco@yahoo.it (E.P.); 2Department of Biomedical Sciences, University of Lausanne, 1005 Lausanne, Switzerland; johannes.burtscher@unil.ch; 3Institute of Sport Sciences, University of Lausanne, 1015 Lausanne, Switzerland; 4Austrian Society for Alpine and High-Altitude Medicine, 6020 Innsbruck, Austria

**Keywords:** winter sports, trauma, sudden death, risk, mortality

## Abstract

Fatalities on ski slopes are very rare, with about one death per one million skier days. Whether the fatality rate is affected by substantial changes in the number of skier days and potentially associated alterations in the structure of the skier population is unknown. Thus, we compared the fatality rate on Austrian ski slopes in the winter season of 2020/21, when skiing activities were dramatically restricted during the COVID-19 lockdown, with those of the previous winter seasons. As a consequence of COVID-19 measures, the number of skier days dropped from over 50 million in previous years to 9.2 million skier days in the winter season of 2020/21. Still, the fatality rate (6.5 deaths/10 million skier days) was not different when compared to any of the seasons from 2011/12 to 2019/20. Despite the lack of international skiers and the reduction in skier days by more than 80%, the fatality rate remained surprisingly unchanged. The weather and snowfall conditions were on average comparable to those of previous winters, and, except for nationality, the composition of the skier population appears to have remained relatively unaltered. In conclusion, the fatality rate during downhill skiing is low and the absolute fatality numbers are primarily a function of the number of skier days.

## 1. Introduction

Millions of downhill skiers practice their favorite sport on snow covered slopes all over the world every year. Undisputedly, this type of winter sports activity contributes to general health benefits related to physical activity but is also associated with a certain risk of injury and even death [1,2]. Downhill skiing-linked risk of death (traumatic and non-traumatic) ranges between 0.5 and 2.0 per million skier days [2,3,4]. According to Tough and Butt (1993) the individuals that are most likely to die from traumatic deaths during downhill skiing are characterized as experienced middle-aged (about 30 years old) male skiers that lose control while skiing too fast, which results in hitting an object and consequential fatal trauma injuries [5]. In contrast, the individuals that are most likely to die from non-traumatic death (typically sudden cardiac deaths) during downhill skiing are represented by elderly (approximately 60 years old) male skiers suffering from one or more cardiovascular risk factors (e.g., prior myocardial infarction) [6,7]. The probability of dying on the ski slope is low and seems to be a rather random event but may be affected by the proportion of skiers with high-risk behavior or afflicted with risk factors predisposing for cardiovascular events. During the last two decades, on average about 50 million skier days have been recorded per winter season in Austrian ski areas and were associated with a relatively stable mortality rate of about 0.7 deaths per 1 million skier days [3].

In the winter seasons from 2000/01 to 2018/19, the number of skier days in Austria ranged from 43.7 (2006/07) to 56.8 million [3,4]. Towards the end of the winter season of 2019/20 (on 11 March in 2020), the World Health Organization (WHO) declared Coronavirus disease 2019 (COVID-19) a global pandemic, resulting in a subsequent restriction of the regular skiing operation in Austria and other Alpine countries. While downhill skiing activities in the winter season of 2019/20 were only moderately affected, a major breakdown occurred in 2020/21. Whether the dramatic drop in the number of downhill skiers is associated with a proportional drop of fatalities is unknown.

Analogous to road accidents, which has recently been elaborated [8], fatal (traumatic) skiing accidents can be attributed to three major potential causes. First, they may be related to the behavior of the skier, e.g., overly-high skiing speed relative to the individuals’ skiing skills, skiing and consuming alcohol, not wearing a helmet, etc. Second, they may be related to the local environment, e.g., weather and snow conditions, slope preparation, etc. Third, they may simply result from (bad) luck, e.g., falling with subsequent head injury on the slope, inculpable colliding with another skier, etc.

It could be argued that the first and third causes were likely to be modified by the pandemic situation and the reduced skier population. Skiers might be more cautious in order to avoid an accident because emergency services and hospitals are occupied with COVID-19 patients, likely associated with a delayed medical treatment. This has been confirmed by reports on diminished sports-related trauma patients during the first lockdown [9]. On the other hand, the risk of colliding with an object or another skier may be reduced due to the lower number of skiers on the slopes [10]. In addition, non-traumatic fatal events (sudden cardiac deaths) may depend on the number of high-risk subjects on the slope [6]. Again, such skiers were expected to be more cautious during the COVID-19 pandemic, and therefore likely to be abstaining from skiing. All of those changes could contribute to potential changes in the mortality rates on ski slopes during the COVID-19 pandemic.

Therefore, the primary aims of the present study were to record the frequency and causes of skiing fatalities in the winter seasons of 2019/20 and 2020/21 and to compare the respective mortality rates with those reported in the years prior to the COVID-19 pandemic.

## 2. Materials and Methods

This brief report is an extension of the study recently published in this journal by Posch et al. (2020) [4]. As described previously, data were collected by members of the Federal Ministry of the Interior (FMI), processed, and stored by the Austrian Board for Alpine Safety (ÖKAS). The incidence of fatalities was calculated based on the number of skier days provided by the Federation of Austrian ski lift companies. This study was performed in conformity with the ethical standards of the 2008 Declaration of Helsinki and was approved (approval ID-62/2019) by the institutional review board (IRB) of the Department of Sport Science, as well as the Board for Ethical Issues (BfEI) of the University of Innsbruck.

### Statistics

Data are presented descriptively (numbers, percentages, and rates). For the calculation of the differences between mortality rates and mortality rate ratios (95% confidence interval and *p*-values) the calculator from MedCalc statistical software (https://medcalc.org, accessed on 8 May 2022) was used, which is based on the guidelines of Sahai and Khurshid (1996) [11]. *p*-values for the differences between mortality rates were obtained by using Chi2-statistic. No correction was performed for multiple testing.

In addition, we provide a table on the mortality rate ratios (95% confidence interval and *p*-value), demonstrating no differences of these ratios in any of the winter seasons from 2011/12 to 2019/20 compared to that of 2020/21. For the confidence interval of the mortality rate ratios, MedCalc uses the “Exact Poisson Method” given on pages 172–174 of Sahai H., Khurshid A. (1996) [11].

## 3. Results

In the winter seasons from 2011/12 to 2019/20, the fatality rate during downhill skiing in Austria varied between 4.6 and 7.9 deaths per 10 million skier days (Figure 1), with an average rate of 6.1 deaths per 10 million skier days. In the winter season of 2020/21 (during the COVID-19 lockdown) the number of skier days dropped from over 50 million in the previous years to 9.2 million skier days, but the fatality rate (6.5 deaths/10 million skier days) was not different when compared to that of any of the seasons from 2011/12 to 2019/20. A significant difference between the fatality rates within this period was only observed between seasons of 2013/14 and 2018/19 (Figure 1).

The causes of death in downhill skiers (on Austrian ski slopes) during the season of the COVID-19 lockdown (2020/21) and the three previous winter seasons are shown in Table 1. The data from the earlier winter seasons have already been reported in this Journal [4]. There were no differences in the distribution of traumatic and non-traumatic deaths. Due to the lockdown in the winter season of 2020/21 and the associated limitations in the entry to and stay in Austrian territory, the skiers were mostly Austrians; therefore, explaining the nationality distribution of the victims. The victims seem to be older in that season when compared to those prior. In Table 2, the mortality rate ratios from 2011/12 to 2019/20 are compared to that of 2020/21. The results do not indicate any differences in the mortality rate ratios between the different winter seasons.

## 4. Discussion

In contrast to our hypothesis, the mortality rate and mortality rate ratios on ski slopes in Austria during the COVID-19 lockdown was not different compared to that of previous winter seasons (2011/12 to 2019/20). Although in the winter season of 2020/21 the number of skier days was reduced by more than 80% and the skier population was mainly composed of Austrian skiers compared to previous seasons, the fatality rate remained stable (6.5/10 million skier days in 2020/21 vs. 6.1/10 million skier days on average in the previous seasons).

This observation indicates that the fatality rates during downhill skiing remained similar to that of previous seasons, despite strongly reduced skier days, the essential lack of non-Austrian skiers, and that only the absolute numbers of rare fatal events primarily depend on the number of skier days. Although we do not have precise data, the structure of the skier population (age and sex distribution) appears to have otherwise been similar to that of the previous winter seasons [12]. Since the local (Austrian) skiers have the opportunity to ski more often than the guests from other countries that have limited access to mountains and ski slopes, they could be expected to have better skiing skills and better knowledge on specific slope conditions, which should provide some prophylactic effect [13]. However, our findings do not support this assumption.

The incidence of skiing injuries (requiring medical treatment) in Austria was reported to be 0.6 per 1000 skier days, accounting for 30,000 ski injuries annually [14], but “only” 1 traumatic death occurred per about 2000 ski injuries (about 50% of all fatalities being traumatic) [3,4]. This means that in only 1 out of about 2000 falls or collisions the resulting injury is severe enough to cause death and thus these cases represent seemingly random events. Certainly, some skiers are more at risk of suffering severe injury and death due to their sex (male), age (young and middle-aged), and behavioral characteristics, such as perceived risk, risk taking, and estimation of ability [15]. The lack of fatalities due to collision with skiers may well be related to the lower frequency—and therefore density—of skiers on the slopes and, therefore, the reduced risk of collision [10].

The other half of fatalities on ski slopes are non-traumatic deaths, predominantly sudden cardiac deaths [7], amounting to approximately three fatalities per ten million skier days [4]. Again, these tragic events are very rare despite the large numbers of elderly downhill skiers suffering from cardiovascular risk factors [3,16]; prior myocardial infarction in male skiers represents the most important risk factor [6]. Other risk factors include systemic hypertension and metabolic diseases, but also low cardiorespiratory fitness due to a lack of specific exercise conditioning [6,7,17,18]. Thus, non-traumatic deaths during downhill skiing are also rare and supposedly random events, although restricted to such high-risk subjects, who might be identified by sophisticated prediction models, which can support effective intervention measures [19]. Our assumption that the high-risk subjects (with regard to cardiovascular adverse events) would rather abstain from skiing during the COVID-19 pandemic seems not to be true, but on the other hand, such subjects may underestimate or simply not be aware of their risks.

As the fatality rate was not affected during the COVID-19 lockdown, despite the dramatically reduced number of skier days, the proportion of skiers at risk may also have remained about the same. The causes of ski injury of the overall skier population and the Austrian skiers are similar [20], and the proportion of downhill skiers in Austria did not considerably change during the past decade, i.e., about 5% are regular, 16% occasional (two to three times per month in the winter season) and 18% rare skiers (one time or less per month) [12]. The weather and snowfall characteristics, which might potentially impact on the fatality risk on ski slopes, also were comparable to previous seasons in 2020/21 [21]. Extreme weather and snowfall conditions could at least partly explain differences in the fatality rates between other winter seasons depicted in Figure 1 and reported by Posch et al. [4]. For instance, the winter season of 2013/14 was characterized by extremely warm temperatures and (in contrast to the southern regions of Austria) by very low snowfall (about 60% less than average) in the northern side of the Alps [21], where most ski areas are. Conversely, the winter season of 2018/19 was characterized by huge volumes of snow (the highest of the existing data recordings [21]). These data support the assumption that high snow cover has some preventive effect on traumatic deaths (e.g., due to better covered rocks and other objects) while less protection by snow cover increases the traumatic fatality risk. This is also confirmed by the extremely large number of fatal traumatic events in the winter season of 2010/11 [4], which was one of the winter seasons with the lowest amounts of snow ever recorded [21].

The reasons for the slightly decreasing trend of traumatic fatality rates during the past decade [4] remains to be elucidated. While the effects of using protective gear, e.g., helmets, are still a matter of controversy [4,22,23,24], rapid emergency medical service by the use of helicopters may furnish at least a partial explanation [3,25,26].

Although the presented unique data allow for the evaluation of a dramatic decrease in the skier population on the fatality rate (on the ski slopes designed for multiple skiers), at least two limitations have to be addressed. First, no precise data on the structures of skier populations (age and sex distributions) in 2020/21 and the previous winter seasons were accessible. Second, information on the number of individuals that are at risk of traumatic and non-traumatic events is extremely difficult to record, further complicating interpretation.

## 5. Conclusions

In conclusion, despite the dramatic reduction in the number of skier days and changes in the composition of the skier population during the COVID-19 lockdown, the mortality rate and mortality rate ratios were not different to those observed during the winter seasons of the previous decade. This indicates that neither behavioral aspects of skiers nor the proportion of subjects at risk of cardiovascular events have considerably changed during the COVID-19 lockdown in the winter season of 2020/21. Thus, skiers may not have been fully aware of the potential consequences (restricted rescue and emergency treatment) of an accident/emergency during the pandemic.

The absolute number of fatalities, but not the fatality rate during downhill skiing, is therefore primarily a function of the number of skier days, provided there are relatively stable distributions/proportions of age, sex, behavioral characteristics, and cardiovascular risk factors within the skier population. Educational work on avoiding risky behavior and medical counseling on the risk of sudden dying, focusing on high-risk groups, remain the primary tools to further reduce fatalities on ski slopes; even more so in times of restricted rescue capacities and high hospital occupancy.

## Figures and Tables

**Figure 1 ijerph-19-07771-f001:**
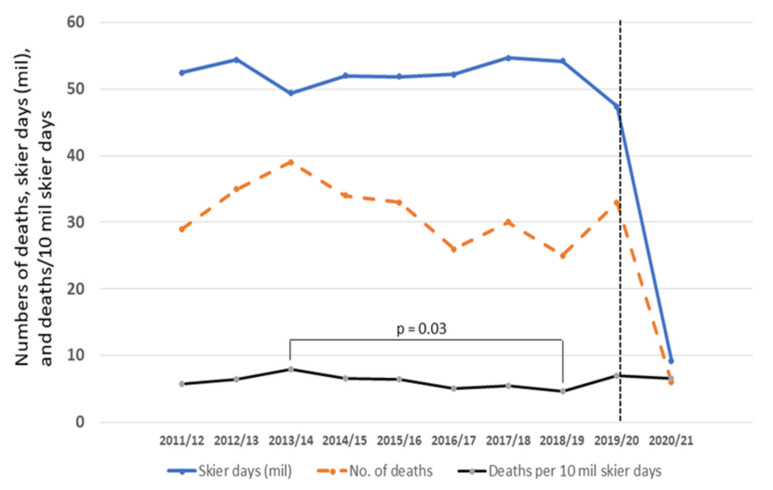
Number of skier days (in millions, blue line), number of deaths during downhill skiing (red line), and fatality rate (deaths/10 million skier days, black line) in the winter seasons of 2011/12 to 2020/21. The dotted vertical line indicates the start of the COVID-19 pandemic (and the subsequent lockdown). Fatality rates were only different between the winter seasons of 2013/14 and 2018/19.

**Table 1 ijerph-19-07771-t001:** Causes of death, sex, age, and nationality distribution of fatalities during downhill skiing in Austria from 2017/18 to 2020/21.

Winter Season	Sex (N) Males/Females	Age (Range) Decade	Nationality (N)	Causes of Deaths	N (%)
2017/18	27/2	2–7	Austrian (12) German (5) Other (12)	Fall	5 (17)
				Collision with object	4 (14)
				Collision with skier	3 (10)
				Sudden non-traumatic death	9 (31)
				Other *	8 (28)
2018/19	22/3	3–8	Austrian (8) German (6) Other (11)	Fall	6 (24)
				Collision with object	3 (12)
				Collision with skier	3 (12)
				Sudden non-traumatic death	7 (28)
				Other *	6 (24)
2019/20	32/1	3–8	Austrian (9) German (14) Other (10)	Fall	7 (21)
				Collision with object	6 (18)
				Collision with skier	1 (3)
				Sudden non-traumatic death	17 (52)
				Other *	2 (6)
2020/21	6/0	6–8	Austrian (6)	Fall	2 (33)
				Collision with object	1 (17)
				Collision with skier	0 (0)
				Sudden non-traumatic death	3 (50)
				Other *	0 (0)

* Other causes of death include avalanche, hypothermia, getting lost, and exhaustion.

**Table 2 ijerph-19-07771-t002:** Mortality rate ratios from 2011/12 to 2019/20 compared to that of 2020/21.

	Skier Days (Millions)	No. of Deaths	Mortality Rate Ratio (95% Confidence Interval; and *p*-Value) (Compared to 2020/21)
2011/12	52.5	29	1.18 (0.401–2.894; 0.69)
2012/13	54.4	35	1.01 (0.349–2.436; 0.94)
2013/14	49.4	39	0.83 (0.286–1.965; 0.70)
2014/15	52.0	34	1.00 (0.342–2.403; 0.97)
2015/16	51.9	33	1.03 (0.351–2.479; 0.92)
2016/17	52.2	26	1.31 (0.441–3.253; 0.54)
2017/18	54.7	30	1.19 (0.405–2.904; 0.67)
2018/19	54.2	25	1.41 (0.474–3.531; 0.44)
2019/20	47.4	33	0.94 (0.321–2.264; 0.92)
2020/21	9.2	6	

## Data Availability

Data sets are stored by the Austrian Board for Alpine Safety (ÖKAS), https://alpinesicherheit.at (accessed on 1 June 2022).

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
