# Peer review of "Unchanged Fatality Rate on Austrian Ski Slopes during the COVID-19 Lockdown"

_ijerph, 2022, doi:10.3390/ijerph19137771_

Round 1

Reviewer 1 Report

The manuscript compares the fatality rate on ski slopes in Austria during the 115 Covid-19 lockdowns and previous winter seasons. 

I would recommend the rejection of a manuscript based on the following:

1- The scientific question trying to answer seems quite naive and not surprising results. 

2-the manuscript does not show any significant contribution to the intended field of research.

3-More importantly, the authors vaguely metioned that the compare the fertality rates between different seasons (lines 74, and 75) using the chi squared tests. Firstly, they did not mentioned how, secondly why they used the plular "tests". Howmany chi-square tests do we have and what tests they are refering to? lastly the chi-squared test does not look an appoporate tool for this purpuse at all. 

Author Response

Ad Reviewer 1:

1- The scientific question trying to answer seems quite naive and not surprising results. 

2-the manuscript does not show any significant contribution to the intended field of research.

3-More importantly, the authors vaguely metioned that the compare the fertality rates between different seasons (lines 74, and 75) using the chi squared tests. Firstly, they did not mentioned how, secondly why they used the plular "tests". Howmany chi-square tests do we have and what tests they are refering to? lastly the chi-squared test does not look an appoporate tool for this purpuse at all. 

Dear Reviewer:

Thank you for reviewing our brief report and your critical suggestions. We agree that the study aims and statistics previously have not been described in sufficient detail. Please, find our responses below which have also been included in the revised version of our manuscript:

ad 1: In our opinion the scientific question is not naïve, although it may seem like it at first glance. Please let us try to provide arguments for this. Fatal events during alpine skiing are very rare events and it is absolutely unknown what happens when the skier population is greatly reduced (by more than 80%) and in case of changes in skiers’ behavior on the slopes, as might have happened during the Covid-19 pandemic. Knowledge on such changes are, however, crucial for adequate policies to prevent skiing accidents.

We added the following paragraphs to the manuscript:

“Analogous to road accidents, for which this has recently been elaborated (Curiel et al., PLoS One 2018; 13(8): e0201890), fatal (traumatic) skiing accidents can be attributed to three major potential causes. First, they may be related to the behavior of the skier, e.g., too high skiing speed related to the individual skiing skills, skiing and consuming alcohol, not wearing a helmet, etc.). Second, they may be related to the local environment, e.g., weather and snow conditions, slope preparation, etc. Third, they may simply result from (bad) luck, e.g., falling with the head on a single stone on the slope, inculpable colliding with another skier, etc.

It could be argued that the first and the third causes were likely modified by the pandemic situation and the reduced skier population. Skiers might be more cautious in order to avoid an accident because emergency services and hospitals are occupied with Covid-19 patients, likely associated with a delayed medical treatment. This has been confirmed by reports on diminished sports-related trauma patients during the first lockdown (van Aert et al., BMJ open 2021). On the other hand, the risk of colliding with an object or another skier may be reduced due to the lower number of skiers on the slopes (Ruedl et al., Sportverletzung/Sportschaden 2013).  

In addition, non-traumatic fatal events (sudden cardiac deaths) depend on the number of high- risk subjects on the slope (Niebauer et al., IJERPH 2021). Again, such skiers are expected to be more cautious during the Covid-19 pandemic, therefore rather abstaining from skiing.”    

Thus, the stable mortality rates on ski slopes during the Covid-19 pandemic is in fact surprising.

As suggest by another reviewer we now do not state a hypothesis and simply present the aims of the study.

ad 2: In light of the responses to (1), we are not sure, why the reviewer thinks that our findings do not provide a significant contribution to the research field. In our opinion, our findings are not only highly novel as no other reports are available on potential changes of the mortality on ski slopes during the Covid-10 pandemic but they also have practical implications: for example, it is of importance to maintain high security standards for skiing activities and consider access restrictions to slopes, even in absence of a high number of potential colliders and even if it could be assumed that skiers behave more responsibly due to high hospital occupancy.

If the reviewer is in possession of existing data on this topic, we would be very grateful if they could share or discuss these data with us.  

ad 3: We agree that the statistics have not been described in sufficient detail. We now provide better information as follows:

“For the calculation of differences between mortality rates and mortality rate ratios (95% confidence interval and p values) the calculator from MedCalc statistical software (medcalc.org, last accessed on 8 May 2022) was used, which is based on the guidelines of Sahai and Khurshid (1996) [11]. P-values for differences between mortality rates are obtained by using Chi2-statistic. No correction was performed for multiple testing.

In addition, we provide a table on the mortality rate ratios (95% Confidence interval and P-value), demonstrating no differences of these ratios in any of the winter seasons from 2011/12 to 2019/20 compared to that of 2020/21. For the confidence interval of the mortality rate ratios, MedCalc uses the "Exact Poisson Method" given on page 172-174 of Sahai H, Khurshid A (1996) [11].”

Ref: Sahai H, Khurshid A (1996) Statistics in epidemiology: methods, techniques, and applications. Boca Raton, FL: CRC Press, Inc.

Reviewer 2 Report

Overall, the topic of this brief report is relevant and should be of interest to the general reader. The authors of the article set the following objective: to examine differences in the fatality rate during downhill skiing in the winter season 2020/21 (Covid-19 lockdown) compared to previous seasons.  The aim of this brief report is well defined, as an extension to a previously published wider study. The introduction part is well crafted and referenced to lead to the aim of the study. The appropriate methodology is used to obtain fatality rates through different skiing seasons. Results are presented clearly, with figures and tables that should be understandable to the general reader. The discussion part is well crafted and is in line with the presented results. Conclusions are in line with the overall facts presented.

References used are up to date, and the overall text is easy to read.

Author Response

Ad Reviewer 2:

Overall, the topic of this brief report is relevant and should be of interest to the general reader. The authors of the article set the following objective: to examine differences in the fatality rate during downhill skiing in the winter season 2020/21 (Covid-19 lockdown) compared to previous seasons.  The aim of this brief report is well defined, as an extension to a previously published wider study. The introduction part is well crafted and referenced to lead to the aim of the study. The appropriate methodology is used to obtain fatality rates through different skiing seasons. Results are presented clearly, with figures and tables that should be understandable to the general reader. The discussion part is well crafted and is in line with the presented results. Conclusions are in line with the overall facts presented.

References used are up to date, and the overall text is easy to read.

Dear Reviewer:

Thank you very much for reviewing our brief report and your favorable assessment!

Reviewer 3 Report

To authors:

This manuscript reported fluctuation of mortality rate on ski slopes from 2011/12 to 2020/21 in Austria. Authors focused on 2020/21 because gross number of skiers was dramatically deduced in the season due to covid-19 lockdown. Authors hypothesized fatality rate would be different from that in other years. As a result, they reported fatality rate in 2020/21 with some information about dead subjects including nationality and cause of death with its number (n), and concluded that fatality rate was proportional to the decreased gross number of skiers.

Reviewer can understand your limitation in the present study. However, still considerable improvement of presentation is needed.

Major:

Reviewer is not sure whether your real purpose is just to fill up one or two plots in 2021/22 year in your series experiment following ref4 (in other words, to know value itself) as “report” or to reveal factor to affect on fatality rate by using specific condition in 2021/22 as “investigation”. In line 59, authors mentioned their hypothesis. I am also not sure whether “hypothesis” is exactly required or not in this kind of study. It depends on what is your purpose.

If authors want to state your hypothesis, it seems difficult to understand why authors thought fatality rate would be different”. In line 24, authors also said “surprisingly”, but the reason why it was surprising is not clear. Additional organization of your logic is needed.

On authors’ logic, the structure of the “ski population”, especially generation (elderly vs. experienced middle age), is a candidate as a factor. How did authors expect that each population is going where? e.g., ratio of elderly population would increase or decrease? If authors expected some changes in the population, authors must mention the expected each vector to lead your hypothesis in the present study. In addition, information about the different fatality rate between elderly and experienced middle age is needed to lead your hypothesis. Authors mentioned a part of these information in Discussion, reviewer strongly recommends that these have to be put before expression your hypothesis.

The following is an example (in the case that elderly population is increased). 1)the fatality rate of elderly population by non-traumatic death is very low, 2)authors expected the increase in elderly population in 2020/21 because…, 3)therefore, authors hypothesized that gross fatality rate in 2020/21 would be decreased. Please show reasonable logic to lead your hypothesis; why “fatality rate would be different”.

Minor:

In Fig. 1, reviewer can imagine authors’ effort to express this figure with three-different units. However, still ingenuity is needed.

In “statistic”, authors mentioned that x2 test was used only for fatality rate. However, in line 93, authors mentioned “victims were older in that season when compared to the previous ones”. Also, in line 89, mentioned “there were no differences in the distribution…”. Did authors do statistical analysis for these comparisons? Please mention it clearly with appropriate expression.

In line 93-95 (Although…), what does “the structure of the skier population” mean? gross population in 2020/21 or population in dead person? This sentence seems to be authors’ private impression. Authors have to put this sentence on “Discussion”, not on “Results”.

In this study, population is an important essence. However, in Table 1, authors showed only range in “Age” column. At least, authors have to provide number (n) in each generation. Also, is there any relationship between generation (e.g., elderly and middle age) and cause of death as reported in ref5? Please give comments in “Discussion”.

In Table 1, could you give some comments about the content of “Other” in cause of deaths, and put it on Discussion.

In line 122, authors mentioned “…despite strongly reduced skier days and the essential lack of non-Austrian skiers…”. Is nationality a factor to affect mortality? If authors think so, please give reasons.

In general, reviewer recommends that authors have to pay more attention to highlight the importance of your study, if this study is not just “record” of 2020/21 season.

Author Response

Ad Reviewer 3:

This manuscript reported fluctuation of mortality rate on ski slopes from 2011/12 to 2020/21 in Austria. Authors focused on 2020/21 because gross number of skiers was dramatically deduced in the season due to covid-19 lockdown. Authors hypothesized fatality rate would be different from that in other years. As a result, they reported fatality rate in 2020/21 with some information about dead subjects including nationality and cause of death with its number (n), and concluded that fatality rate was proportional to the decreased gross number of skiers.

Reviewer can understand your limitation in the present study. However, still considerable improvement of presentation is needed.

Dear Reviewer:

Thank you very much for reviewing our brief report and your helpful and constructive suggestions. We tried to address adequately all your comments in the point-to-point responses below and revised our manuscript accordingly.

Major:

Reviewer is not sure whether your real purpose is just to fill up one or two plots in 2021/22 year in your series experiment following ref4 (in other words, to know value itself) as “report” or to reveal factor to affect on fatality rate by using specific condition in 2021/22 as “investigation”. In line 59, authors mentioned their hypothesis. I am also not sure whether “hypothesis” is exactly required or not in this kind of study. It depends on what is your purpose.

If authors want to state your hypothesis, it seems difficult to understand why authors thought “fatality rate would be different”. In line 24, authors also said “surprisingly”, but the reason why it was surprising is not clear. Additional organization of your logic is needed.

Re:  We agree that the study aims have not been described in sufficient detail. The primary study goal was simply to assess what happened when the skier population was greatly reduced (by more than 80%) and the skiers’ behavior on slopes may have substantially changed during the Covid-19 pandemic. We also agree that it is not necessary to state hypotheses. Instead, we elaborated on the rationale of the study and the study aims (please, see below). 

On authors’ logic, the structure of the “ski population”, especially generation (elderly vs. experienced middle age), is a candidate as a factor. How did authors expect that each population is going where? e.g., ratio of elderly population would increase or decrease? If authors expected some changes in the population, authors must mention the expected each vector to lead your hypothesis in the present study. In addition, information about the different fatality rate between elderly and experienced middle age is needed to lead your hypothesis. Authors mentioned a part of these information in Discussion, reviewer strongly recommends that these have to be put before expression your hypothesis. The following is an example (in the case that elderly population is increased). 1)the fatality rate of elderly population by non-traumatic death is very low, 2) authors expected the increase in elderly population in 2020/21 because…, 3) therefore, authors hypothesized that gross fatality rate in 2020/21 would be decreased. Please show reasonable logic to lead your hypothesis; why “fatality rate would be different”.

Re: Thank you for your suggestions. We do no longer state a hypothesis but describe in the introduction section more clearly now why we expected potential changes in the fatality rate:

“Analogous to road accidents, for which this has recently been elaborated (Curiel et al., PLoS One 2018; 13(8): e0201890), fatal (traumatic) skiing accidents can be attributed to three major potential causes. First, they may be related to the behavior of the skier, e.g., too high skiing speed related to the individual skiing skills, skiing and consuming alcohol, not wearing a helmet, etc.). Second, they may be related to the local environment, e.g., weather and snow conditions, slope preparation, etc. Third, they may simply result from (bad) luck, e.g., falling with the head on a single stone on the slope, inculpable colliding with another skier, etc.

It could be argued that the first and the third causes were likely modified by the pandemic situation and the reduced skier population. Skiers might be more cautious in order to avoid an accident because emergency services and hospitals are occupied with Covid-19 patients, likely associated with a delayed medical treatment. This has been confirmed by reports on diminished sports-related trauma patients during the first lockdown (van Aert et al., BMJ open 2021). On the other hand, the risk of colliding with an object or another skier may be reduced due to the lower number of skiers on the slopes (Ruedl et al., Sportverletzung/Sportschaden 2013). 

 Thus, the stable mortality rates on ski slopes during the Covid-19 pandemic is actually surprising (at least in our opinion). 

We also mention:

“In addition, non-traumatic fatal events (sudden cardiac deaths) may depend on the number of high- risk subjects on the slope (Burtscher et al., Int J SportsMed 2000). Again, such skiers are expected to be more cautious during the Covid-19 pandemic, therefore rather abstaining from skiing.”

Minor:

In Fig. 1, reviewer can imagine authors’ effort to express this figure with three-different units. However, still ingenuity is needed.

Re: Thank you. We now explain the numbers by adding the respective line color in the legend.

In “statistic”, authors mentioned that x2 test was used only for fatality rate. However, in line 93, authors mentioned “victims were older in that season when compared to the previous ones”. Also, in line 89, mentioned “there were no differences in the distribution…”. Did authors do statistical analysis for these comparisons? Please mention it clearly with appropriate expression.

Re: Thank you, we agree. We now describe the statistics (with regard to mortality rates) in more detail. Other aspects (e.g., age, etc. are only presented descriptively). Ad statistics we added as follows: 

“For the calculation of differences between mortality rates and mortality rate ratios (95% confidence interval and p values) the calculator from MedCalc statistical software (medcalc.org, last accessed on 8 May 2022) was used, which is based on the guidelines of Sahai and Khurshid (1996) [11]. P-values for differences between mortality rates are obtained by using Chi2-statistic. No correction was performed for multiple testing.

In addition, we provide a table on the mortality rate ratios (95% Confidence interval and P-value), demonstrating no differences of these ratios in any of the winter seasons from 2011/12 to 2019/20 compared to that of 2020/21. For the confidence interval of the mortality rate ratios, MedCalc uses the "Exact Poisson Method" given on page 172-174 of Sahai H, Khurshid A (1996) [11].”

In line 93-95 (Although…), what does “the structure of the skier population” mean? gross population in 2020/21 or population in dead person? This sentence seems to be authors’ private impression. Authors have to put this sentence on “Discussion”, not on “Results”.

Re: Thanks again. We added “age and sex distribution” and moved this sentence into the discussion section.  

In this study, population is an important essence. However, in Table 1, authors showed only range in “Age” column. At least, authors have to provide number (n) in each generation. Also, is there any relationship between generation (e.g., elderly and middle age) and cause of death as reported in ref5? Please give comments in “Discussion”.

Re: Due to the low numbers we only provide this rough information but now comment on that (as suggested) in the discussion section.  

In Table 1, could you give some comments about the content of “Other” in cause of deaths, and put it on Discussion.

Re: We now provide the following information on “other causes of death” in the table legend:

*Other causes of death include avalanche burial, hypothermia, getting lost and subsequent exhaustion.

In line 122, authors mentioned “…despite strongly reduced skier days and the essential lack of non-Austrian skiers…”. Is nationality a factor to affect mortality? If authors think so, please give reasons.

Re:  Thank you, this is a good point. Local (Austrian) skiers have the opportunity to ski more often than guests from other countries that have limited access to mountains and ski slopes. Therefore, Austrian skiers are expected to have better skiing skills and better knowledge on specific slope conditions which should provide some prophylactic effect. However, our findings do not support this assumption. This aspect has now been added in the discussion section.    

In general, reviewer recommends that authors have to pay more attention to highlight the importance of your study, if this study is not just “record” of 2020/21 season.

Re: Thank you very much again. We are convinced that based on your concerns/comments we could substantially improve our brief report overall and specifically better highlight the importance and implications.

Round 2

Reviewer 1 Report

Thanks for your answers and clarifications.

Reviewer 3 Report

Reviewer is grateful for authors’ considerable effort to revise. The manuscript, especially Introduction, has been successful to highlight your purpose, experimental meaning, and your logic. And authors responded to all of reviewer’s requests. Although scientific sounds are still not very high, given this is in “brief report”, this paper would provide some interesting points to be thought. Now nothing to say for authors. Thank you for your cooperation.